# Internalized Racism and Mental Health: The Moderating Role of Collective Racial Self-Esteem

**DOI:** 10.3390/bs14111003

**Published:** 2024-10-29

**Authors:** Steven M. Sanders, Tiffany R. Williams, April T. Berry, Claudia Garcia-Aguilera, Kiera Robinson, Reniece Martin, Paigean Jones

**Affiliations:** 1School of Psychological Science, Oregon State University, Corvallis, OR 97331, USA; 2School of Medicine, Indiana University, Indianapolis, IN 46202, USA; tw84@iu.edu; 3College of Education & Professional Studies, University of South Alabama, Mobile, AL 36688, USA; aberry@southalabama.edu; 4Department of Psychological Sciences and Counseling, Tennessee State University, Nashville, TN 37209, USA

**Keywords:** internalized racism, psychology, depression, anxiety, moderation

## Abstract

Internalized racism is the internalization of beliefs about racism and colonization that contribute to the acceptance of negative messaging or stereotypical misrepresentations that inform perceptions about worth and ability. Internalized racism is associated with psychological distress in racially diverse people. Collective racial self-esteem is a potential protective factor that can serve as a moderator in reducing distress and facilitating psychological well-being. The sample for the present study consisted of 526 participants who self-identified as African American, Asian American/Pacific Islander, Latinx American, or American of Arab or Middle Eastern descent. The researchers used regression with the Process macro to investigate the potential moderating properties of collective racial self-esteem on the relationship between internalized racism and psychological distress in the sample. The findings indicated that specific domains of collective racial self-esteem moderated the internalized racism–psychological distress relationship.

## 1. Introduction

Internalized racism, or appropriated racial oppression, is a phenomenon that people of color (POC) experience when they adopt aspects of racism by accepting negative messaging about their worth and abilities based on their ethnic or racial group membership [1,2,3]. In other words, “the oppressed will identify and emulate the oppressive actions of their oppressor” [4]. Campón and Carter (2015) [1] theorized a model of internalized racism comprising four dimensions: emotional responses, American standards of beauty, devaluation of one’s own group, and patterns of thinking. Emotional responses refer to the experience of strong negative emotions related to one’s own racial group (e.g., shame, self-hatred, or guilt; [5]). American standards of beauty are characterized as one who embraces Eurocentric aesthetic preferences (e.g., lily complex, thinness ideal; [6]). Devaluation of one’s own group refers to internalized negative beliefs a person has about their own racial group (e.g., people of my race do not have much to be proud of; [1]). Lastly, patterns of thinking are described as the embracing of beliefs that sustain the societal status quo (e.g., people take racial jokes too seriously; [1]).

When POC adopt oppressive messages about their ethnoracial identities, perpetuated in the White dominant culture and informed by Eurocentric ideologies and colonization, the internalization of these messages can contribute to self-loathing or worse mental health (i.e., depression and anxiety; [4,7,8]). Internalized racism is evidenced to be linked with several psychological conditions, such as feelings of hopelessness and stress, symptoms of depression or anxiety, and poor perceptions of body image [4]. Internalized racism was found to have a negative association with self-esteem, life satisfaction, and psychological well-being. Although internalized racism is a risk factor for mental health outcomes in a range of marginalized populations, approximately 40% of the research focuses on the experiences of African Americans [4]. However, many other ethnoracial groups also experience the negative impacts of internalized racism.

Though the previous literature focuses on the internal effects of discrimination and racism in relation to racial identity and other protective factors specifically among the African American population, little research has examined the psychological processes of internalized racism for other ethnoracial identities (e.g., Asian Americans/Pacific Islanders, Latinx Americans, and Arab/Middle Eastern/North African descent). Research on racial identity development provides information about how those with ethnoracial identities may identify but do little in investigating internalized racial attitudes for non-White populations [1]. Thus, the goal of this present study was to understand additional concepts such as collective racial self-esteem (i.e., a person’s self-concept as it relates to membership in a specific social group; [9,10]) and how it enhances the relationship between internalized racial attitudes and adverse mental health outcomes for those with ethnoracial identities. Identifying such relationships can inform the practice for mental health professionals in assisting those with these life experiences and assist with combating negative outcomes that may be associated with internalized racism in individuals with marginalized ethnoracial identities. Throughout this paper, various abbreviations and naming conventions are used to refer to specific racial groups (e.g., Black versus African American versus AA; Asian American/Pacific Islander versus AAPI versus Asian). We are describing the same groups of people throughout the paper regardless of the naming convention employed.

### 1.1. Racial Minority Individuals’ Experiences of Internalized Racism

#### 1.1.1. African Americans

Internalized racism experienced by African Americans (AAs) affects their socialization, racial or ethnic identity, relational development, perceptions of self, and quality of life [11]. Bailey and colleagues (2011) theorized a model to describe internalized racism within the context of Black racial identity comprising five categories: (1) internalization of negative stereotypes, (2) self-destructive behaviors, (3) devaluation of the African worldview and motifs, (4) belief in the biased representation of history, and (5) alteration of physical appearance [5]. First, internalization of negative stereotypes refers to one’s acceptance of racialized stereotypical messaging or images about AA people and is described as having a two-fold effect: AAs will (a) deny or reject an African-centric cultural worldview and (2) accept and internalize Eurocentrism rooted in colonization, racist stereotypical messaging, discriminating or oppressive doctrines and systems, or White supremacist worldviews or colonial mentality or beliefs. Second, self-destructive behaviors are described as harmful or threatening actions toward the self and others in the AA community thwarting functioning and survival. Third, belief in a biased representation of history refers to the endorsement of misrepresentations of history that privilege White supremacy and colonization, while denying the marginalization of AAs and negative effects of racist practices and policies such as chattel slavery in the Americas [12,13,14]. Fourth, the devaluation of the African worldview and motifs is characterized as dismissal of African ideologies, cultures, and values, such as rejecting collectivism or communal harmony, honoring of elders, and egalitarianism [5,15]. Lastly, alteration of physical appearance refers to AAs who transform their outward appearance to features that align more closely with Eurocentric aesthetic principles. Internalized racism affects AAs in insidious ways, as it can occur consciously or unconsciously, and can have deleterious effects on AAs.

Previous research on mental distress among AAs demonstrates a positive association between internalized racism and several psychological factors including depression [16,17], anxiety [8], social anxiety [18], self-esteem [19,20], stress [21], and sleep disturbance [22]. The literature notes a shift in how internalized racism is understood, specifically, shifting the focus from solely measuring internalized racism through negative stereotypical messaging endorsed by society to (a) the value of one’s racial identity and (b) examining modifications of appearance that are more aligned with Eurocentric standards [5]. Western culture and Eurocentric values are steeped in the United States of America’s mainstream culture and systems. Thus, naturally, internalized racism has a measurable negative effect on AAs’ livelihoods, career aspirations, and psychological health [5,23].

#### 1.1.2. Asian Americans and Pacific Islanders

There is a growing body of evidence on the impacts of internalized racism among Asian Americans and Pacific Islanders (AAPIs). By no means are AAPIs a homogenous group, but it is a term used to be inclusive in representing the array of diverse individuals that identify with this community. Like AAs, AAPIs were also affected by European colonization, systemic oppression, discrimination, hate, racist legislation (e.g., Japanese internment during WWII), and harassment [12,24,25]. Internalized racism among AAPIs can be viewed as an adaptive mechanism in that assimilation into Eurocentric culture or occupying White spaces may help them feel less devalued and more capable of being successful in the United States of America [24,26].

Several characteristics comprise the experience of internalized racism for AAPIs. For example, believing messages of racial inferiority, self-devaluation, and embodying a colonial mentality are all directly associated with the lengthy history of colonization and misappropriation of Asian cultures [26,27]. Common stereotypes that AAPIs battle include model minority, perpetual foreigner, and gender-specific or beauty misconceptions [25]. Internalized racism is evidenced to contribute to poor physical [21] and psychological distress [28] for AAPIs, for example, psychological distress and social isolation [26] as well as depression and anxiety [1,29,30]. Additionally, low self-esteem is associated with social- or race-specific stereotypical misrepresentations of Asian Americans [26,31].

#### 1.1.3. Latinx Americans

Research examining the relationship between internalized racism and Latinx individuals is especially sparse. The shift in the sociopolitical climate during the presidency of Donald Trump with the implementation of harmful immigrant policies reflects the perpetuation and the prevalence of discrimination toward Latinx communities [32]. Experiences of internalized racism often intersect with multiple systemic oppressions that impact the well-being and quality of life of racially marginalized communities [17,32]. The United States of America’s policies targeting Latinx communities and rooted in colonialism (e.g., the Platt amendment, mass expulsion, state-sanctioned violence, or the Monroe doctrine) perpetuate systemic oppression and devaluation of Latinx people [33,34,35].

Some researchers argue that stereotypical messages about racial inferiority (e.g., the myth of meritocracy) are concealed in the United States of America’s propaganda, media, and systems (e.g., the biased and unjust judicial system). Thus, people in Latinx communities may be more likely to accept dominant culture or Eurocentric ideologies, internalize oppressive lived experiences, and develop learned helplessness, perpetuating internalized racism [35,36]. Previous research demonstrates that experiences of internalized racism among Latinx communities include an array of factors, such as self-hatred and self-doubt, assimilation or acculturation, negative stereotypes, and racial identity [35,36,37]. Mental, physical, and emotional strain can generate profound psychological distress for Latinx people navigating internalized racism [36,38]. As mentioned, the combination of internalization of negative stereotypes, assimilation to Eurocentric ideologies, and overcoming barriers can be stressful and taxing. Cultural barriers Latinx people navigate include skin color, stereotypes, and language, which serve as deterrents to acceptance and fuel for discrimination [38].

#### 1.1.4. Americans of Arab or Middle Eastern Descent

Even less is known about experiences of internalized racism among Americans of Arab or Middle Eastern/North African (MENA) descent. As a racial minority group who are labeled as a “White” racial category on the United States of America’s census and “other”, Arab/MENA people are also impacted by the racial landscape of America and colonization [39]. In fact, self-identification of Arab/MENA persons is far more complicated than identifying with racial or ethnic categories but is inclusive of ethnoracial markers, such as cultural or ancestral traits or religion and names [40]. Prejudice, invisibility, and discrimination shape Arab/MENA communities’ lived experiences, which are especially heightened after the 9/11 terrorist attacks [39].

Multilevel oppressions at the individual (e.g., microaggressions), institutional (e.g., social exclusion in the workforce), and societal (e.g., policies like the USA PATRIOT Act) levels complicate Arab/MENA quality of life and existence [32]. In a study that examined the impact of acculturation, racial identity, and religious affiliation among Arab/MENA people, Awad (2010) [39] found that 59% of participants indicated awareness of denigrating stereotypes about their race, and 77% experienced overt forms of racial discrimination. The implications of racial profiling, stigma, social marginalization, acculturative stress, and discrimination have been examined in Arab Americans [41], but there are no studies specifically examining internalized racism in participants of Arab descent. The health impacts of internalized racism remain unknown; however, it is likely that internalized racism has an adverse effect on the psychological well-being of Arab/MENA people.

### 1.2. Collective Racial Self-Esteem

To understand collective racial self-esteem, collective self-esteem must be understood first. Collective self-esteem was first studied by Luhtanen and Crocker (1992) when they believed there was a relationship between individuals’ self-esteem and how they felt about the groups they were members of. It has been defined as the component of an individual’s self-image that develops from their relationships and interactions with others and their group membership(s) [42]. To further understand this concept, Crocker and Luhtanen (1990) [43] developed a scale to measure collective self-esteem, exploring four broad domains: (1) private collective self-esteem (i.e., positive evaluation of one’s group), (2) membership esteem (i.e., how one sees oneself in a group; are they a good member?), (3) public collective self-esteem (i.e., how the group one belongs to is evaluated by others), and (4) importance to identity (i.e., how important membership in a group is to self-concept) [42]. Each domain assesses individuals’ social identity based on their membership in ascribed groups to include race, ethnicity, and gender and can serve as protective factors against mental health outcomes such as depression and anxiety [43].

In the context of cross-cultural research, collective self-esteem has been further explored through the construct of collective racial self-esteem. Research has found that collective racial self-esteem acts as a protective factor against the effects of racism [44,45,46,47]. Collective racial self-esteem refers to a person’s self-concept and evaluation of the self, as it relates to membership in a specific racial group [9,10]. A positive perception of one’s group and membership within that group is linked to increased psychosocial well-being among racially diverse populations [48]. There is empirical evidence demonstrating that positive collective racial self-esteem is related to decreased symptoms of depression and anxiety, increased utilization of positive coping skills, and improved quality of life [44,45,46,47,49].

Higher collective racial self-esteem has been shown to be a mitigating factor that reduces symptoms of depression and anxiety among immigrant populations [46,50,51]. Another study found that Latinx adolescents with higher levels of collective racial self-esteem related to their culture were less likely to experience depression and anxiety as compared to those who had low levels [50]. Similar results were found in a study that examined the impact of collective racial self-esteem and racial identity on the quality of life of Colombian immigrants residing in Chile [51]. The findings suggested that collective racial self-esteem improved quality of life and decreased symptoms of depression and anxiety [51]. Among AA adolescent girls, collective racial self-esteem was shown to decrease the impact of race-based stressors [52,53]. Furthermore, it was found to decrease symptoms of depression and anxiety among AA boys and girls [53]. Thus, collective racial self-esteem has the potential to serve as a moderating or buffering factor in facilitating psychological well-being for racially diverse people.

### 1.3. The Present Study

Previous research has consistently linked internalized racism to adverse psychological outcomes. There is strong empirical evidence for the negative psychological associations with internalized racism for Black individuals. However, there is still room in the literature to examine this relationship for diverse populations (i.e., other racial groups) and to examine potential moderating constructs. The goal of the present study was to re-examine the relationship between internalized racism and depression and anxiety (see Figure 1) and explore the potential moderating role of collective racial self-esteem, including its four domains (see Figure 2). The present study is one of the first to examine the relationships between internalized racism and psychological distress (i.e., depression and anxiety) for people who self-identify as Arab/MENA. The findings also add to a growing body of research that investigates the psychological impacts of internalized racism and the protective effects of collective self-esteem for people who self-identify as AA, AAPI, and Latinx.

## 2. Materials and Methods

### 2.1. Participants

Participants included 538 individuals who identified as racial minorities whose ages ranged from 18 to 65 years old (M = 29.87, SD = 9.11; see Table 1) and were recruited by the researchers using Amazon Mechanical Turk (MTurk). The sample included 163 AAs (30.3%), 201 AAPIs (37.4%), 96 Latinx Americans (17.8%), 68 Arab/MENA people (12.6%), 9 Alaskan Native people (1.7%), and 1 multiracial person (0.2%). Additionally, 248 participants (46.1%) identified as male, 283 participants (52.6%) identified as female, 4 participants (0.7%) identified as transgender, and 3 participants (0.6%) identified as gender non-binary. Finally, 171 participants (31.8%) reported having an annual income above USD 60,000, and 294 participants (54.7%) reported being in a relationship (i.e., married or dating exclusively). To be eligible to participate in the study, individuals had to be over the age of 18, identify as a racial/ethnic minority person, and live in the United States of America or its territories. All participants provided informed consent, and the [institution redacted] institutional review board approved the study and its protocol. Additionally, a G*Power post hoc power analysis with an input effect size of 0.10 indicated an achieved power of 0.825, a power level that suggested a high probability of correctly rejecting the null hypothesis [54].

### 2.2. Measures

#### 2.2.1. Internalized Racism

The Appropriated Racial Oppression Scale (AROS; [1]) is a 24-item measure that assesses the degree to which racial minorities internalize racist experiences and media. Items use seven-point strength-of-agreement Likert scales with responses ranging from 1 (strongly disagree) to 7 (strongly agree) and aligned with Campón and Carter’s four domains of internalized racism: emotional responses (7 items; e.g., “People take racial jokes too seriously”), American standard of beauty (6 items; e.g., “Good hair” (i.e., straight) is better”), devaluation of own group (8 items; e.g., “People of my race don’t have much to be proud of”), and patterns of thinking (3 items; e.g., “People take racial jokes too seriously”). Previous research has used this instrument to measure internalized racism in participants of color [55]. The present study used the total score of the AROS-24 (i.e., the sum of responses for all 24 items) for all analyses, which ranged from 24 to 168, with higher scores indicative of higher levels of internalized racism. The AROS-24 has an observed reliability coefficient that was more than acceptable (α = 0.96).

#### 2.2.2. Psychological Distress

##### Depression and Anxiety

Depression and anxiety were measured using the Depression Anxiety and Stress Scale 21-item short form (DASS-21; [56]). Researchers developed the DASS in response to criticisms that the Beck Depression Inventory (BDI) and the Beck Anxiety Inventory (BAI) lacked a satisfactory level of discriminant validity, as there are overlapping measures of depression and anxiety due to the overlapping commonly shared symptoms and shared root of negative affectivity [56]. The depression and anxiety subscales of the DASS-21 each include seven items rated on four-point frequency Likert scales ranging from 0 (did not apply to me at all—NEVER) to 3 (applied to me very much, or most of the time—ALMOST ALWAYS). Sample items for the depression subscale include “I couldn’t seem to experience any positive feeling at all” and “I found it difficult work up the initiative to do things”. Sample items for the anxiety subscale include “I was aware of dryness of my mouth” and “I felt I was close to panic”. The scores range from 0 to 42, with the raw scores ranging from 0 to 21 and being multiplied by two, with higher scores indicating more endorsements of depressive or anxious symptoms and mood. For depression, scores ranging from 0 to 9 are categorized as “normal”, 10 to 12 as “mild”, 13 to 20 as “moderate”, 21–27 as “severe”, and 28 and greater as “extremely severe”. For anxiety, scores ranging from 0 to 6 are categorized as “normal”, 7 to 9 as “mild”, 10 to 14 as “moderate”, 15–19 as “severe”, and 20 and greater as “extremely severe”. Previous research found strong support of the DASS-21’s internal consistency, with the depression and anxiety subscales having Cronbach’s alpha coefficients ranging from 0.88 to 0.90 and 0.82 to 0.85, respectively [57,58,59]. In the present study’s sample, the depression (α = 0.92) and anxiety (α = 0.89) subscales had internal reliability coefficients that were more than acceptable.

#### 2.2.3. Collective Self-Esteem

Collective racial self-esteem, or self-esteem associated with one’s racial group, was assessed using the 16-item Collective Self-Esteem Scale-Race Specific Form (CSES; [42]). Items use a 7-point strength-of-agreement Likert scale with responses ranging from 1 (strongly disagree) to 7 (strongly agree). This instrument assesses four domains of racial self-esteem, including membership self-esteem (i.e., how good or worthy a member of the racial group one is; e.g., “I am a worthy member of my race/ethnic group”), private collective self-esteem (i.e., how good one’s social groups are; e.g., “I often regret that I belong to my racial/ethnic group”), public collective self-esteem (i.e., how one believes others evaluate one’s social groups; e.g., “Overall, my racial/ethnic group is considered good by others”), and importance to identity (i.e., how important one’s group is to one’s self concept; e.g., “Overall, my race/ethnicity has very little to do with how I feel about myself”). Although it is possible to create a total score for collective self-esteem, the authors discourage the practice for psychometric reasons. Specifically, the authors suggest the potential for misleading or inaccurate findings if an aggregate scale score approach is used by researchers because certain subscales are not correlated. Scores on each domain range from 4 to 28, with higher scores indicating higher levels of racial collective self-esteem. In the present study’s sample, the membership self-esteem (α = 0.82), private collective self-esteem (α = 0.81), public collective self-esteem (α = 0.80), and importance to identity (α = 0.81) domains of the CSES had internal reliability coefficients that were acceptable.

#### 2.2.4. Demographics

Demographics were gathered using a researcher-designed demographic questionnaire. This questionnaire included age (open-ended item with answer validation requiring a number but not requiring a response), gender identity (e.g., man, woman, gender non-binary, or other), educational attainment (e.g., high school diploma, some college, etc.), household income categories (e.g., USD 10,000–30,000, USD 30,001–50,000, or USD 50,001–70,000), employment status (e.g., student, unemployed, employed part-time, etc.), relationship status (e.g., single, in a dating/committed relationship, etc.), religion (open-ended response item), veteran status (binary response item with yes/no option), racial identity (e.g., Black/African American, White, Asian, Pacific Islander, Arab, Hispanic/Latinx, or multiracial), and sexual orientation (e.g., straight, gay, or lesbian). Participants were able to self-identify their racial category and were able to choose as many as they felt were applicable to themselves. However, as there was only one participant who identified as multiracial, we did not include them in our analyses.

### 2.3. Procedures

After receiving approval from the host institution IRB, participants were recruited through the MTurk platform. MTurk is a crowdsourcing platform designed for using people to complete various tasks online [60]. MTurk provides researchers with the ability to gather large samples for social science in a relatively inexpensive manner [61]. Additionally, the data gathered using MTurk are high-quality, with high levels of internal consistency compared to data collected using different methods [61]. Another benefit of using MTurk is that the population is generally more diverse than the typical sample of college students used in social science research [60]. Sheehan posits that using MTurk avoids the potential biases of sampling college students. Social science researchers use several techniques to ensure the quality of data collected from MTurk, including using attention checks, screening participants to ensure they meet inclusion criteria and limiting the amount of time a worker has to complete the survey. Research has suggested that at any given time, researchers have access to between 10,000 and 100,000 potential research participants [62].

Previous research found a statistically significant difference in results between MTurk participants and participants recruited using more traditional methods, but those differences lacked practical significance [63]. Bartneck and colleagues suggest that difference is due to the MTurk sample’s increased diversity compared to the sample gathered on a college campus. Additional research found that data gathered from an MTurk sample replicated findings from previous research more closely than data collected in person [64]. All participants were administered a survey using Qualtrics. Attention checks were utilized where participants were instructed to “Choose Strongly Agree for this item”, and participants who failed to choose the appropriate option were removed from the final dataset for quality assurance of data. There were two such items in the survey instrument. The MTurk participants were given a maximum of 50 min to complete the survey instrument, as required at the time by MTurk policies, ensuring that automated systems were not being used to respond to the survey instrument. The participants were administered the survey instrument via Qualtrics after being provided informed consent and were instantly compensated USD 0.60 for participating. The survey instrument was only presented in English. The data were collected from December 2019 to January 2020.

### 2.4. Research Design

The researchers used a correlational research design to answer the research questions associated with the study. Correlational designs are an approach used to explore the relationships between at least two variables [65]. Additionally, the present study tested the moderation properties of the various domains of collective self-esteem on the internalized racism–psychological distress relationship. Moderator variables affect the strength and/or direction of the relationship between an independent and dependent variable [65].

### 2.5. Data Analysis

To examine the relationships between internalized racism and psychological distress in all participants, linear regression was used in SPSS version 29. To examine the moderation properties of collective racial self-esteem, linear regression with the Hayes PROCESS macro 4.2 was used [66]. The PROCESS macro is a statistical computational aid for researchers estimating regression models, incorporating mediation and moderation options. The PROCESS macro is integrated into the SPSS version 29 regression menu. Additionally, when prompted, the PROCESS model generates SPSS code to plot moderation analysis results. The PROCESS macro examines interactions between the predictor and proposed moderator(s) to determine whether the predictor variable has an effect on the dependent variable that varies based on the value of the moderator variable.

### 2.6. Researcher Positionality Statement

The identities of researchers and their role(s) in research projects can impact the research process and interpretations of data [67]. In qualitative research, positionality is the declaration and recognition of one’s own position in a piece of academic work [68]. Although this work is quantitative in nature, the recruitment of participants based on their identities *requires* the research team to disclose relevant details about their own personal and scholarly positionality to avoid the appearance of bias(es) [69]. The lead author is a cisgender Black man trained in counseling psychology and is an assistant professor of psychological science. As part of his doctoral experience, he received training in advanced statistical analyses and approaches, working with marginalized populations, and psychological assessment. His research and clinical interests include physical and mental health disparities in marginalized populations, internalized racism, identity development, and coping and adjustment strategies used by college students. The second author identifies as a Black heterosexual woman who grew up in the Midwest. She was a first-generation college student and was raised in a low socioeconomic background. She has a doctoral degree in urban education specializing in counseling psychology. She serves as faculty and enjoys mentoring students, especially if they identify as underrepresented racial minorities. She is licensed as a psychologist–health service professional designation in Ohio, Tennessee, and Indiana. She serves as a supervisor to diverse psychologists-in-training to increase the number of culturally sensitive and competent professionals in psychology. Her research and clinical interests include investigating health disparities, gendered racism, systemic racism, and impacts on minority mental health. The third author is a cisgender Black woman, counseling psychologist, and has more than 12 years of research experience on the racialized experiences of marginalized people. The fourth author is a cisgender Hispanic woman who is an advanced undergraduate research student. Her research interests include gender identity development, racism, and coping strategies. The fifth author is a cisgender biracial woman who is an advanced undergraduate research student. Her research interests include racism, racial identity development, and psychological adjustment. The sixth author is a Black woman living in America, a wife, and mother of two children. She is a doctoral candidate in a counseling psychology Ph.D. program who holds a master’s degree in psychology. She is a current predoctoral intern who is passionate about working with individuals struggling with severe mental illness (SMI) and all intersections of at-risk populations (e.g., at-risk youth, SMI, prison, and restorative justice). Her research interests include examining decolonial healing, spirituality, restorative justice, and protective factors that promote well-being for people in Black communities. Finally, the seventh author is a Black woman, born in a small town in the rural south, who hails from a family the majority of whom hovered around the federal poverty line. She was not taught self-care by her family, nor did she have a familial model of self-care behaviors. She earned a doctoral degree in social work, and in her family, hard work was not only required for survival but was seen as a badge of honor. Her research and clinical interests include holistic medicine, Black perinatal and postpartum mental health, and improving the health and wellness of Black women. Collectively, the research team’s breadth and depth of training, experiences, and expertise in psychology made them well-equipped to examine the negative effects of internalized racism and the moderating properties of collective racial self-esteem.

## 3. Results

### 3.1. Direct Effects of Internalized Racism on Mental Health Outcomes

The linear regression analysis showed that internalized racism was a statistically significant predictor of depression for the sample at large and Black, Asian, Hispanic, and Arab participants specifically (see Table 2). Regarding anxiety, analyses indicated that internalized racism was a statistically significant predictor for the sample at large and Black, Asian, and Hispanic participants specifically (see Table 3), but this relationship was statistically insignificant for Arab participants. These findings indicate that as the internalization of racism increases, so does the endorsement of depressive and anxious symptomatology for all participants, except for Arab participants and anxiety.

### 3.2. Collective Self-Esteem as a Moderator of the Internalized Racism and Mental Health Outcomes Relationship

To test the hypothesis that domains of collective self-esteem moderate or modify the direction and/or magnitude of the relationship between internalized racism and psychological distress, the Hayes PROCESS macro’s Model 1 was used. The results below are organized by racial group and follow the same reporting pattern as the direct relationships above (i.e., entire sample followed by various racial groups). There were separate models estimated to test each moderator one at a time. Regarding the domains of collective self-esteem that moderate the internalized racism and depression relationship for the sample at large, importance to identity (*B* = −0.024, *SE* = 0.011, *p* = 0.032), membership collective self-esteem (*B* = −0.030, *SE* = 0.011, *p* = 0.006), and private collective self-esteem (*B* = −0.028, *SE* = 0.011, *p* = 0.008) were all statistically significant moderators. These negative coefficients demonstrate the buffering effects of collective self-esteem on the internalized racism–depression association, meaning that as collective self-esteem increased, the consequences of internalized racism (i.e., depressive symptoms) were reduced. Regarding the domains of collective self-esteem that moderate the internalized racism and anxiety relationship for the sample at large, importance to identity (*B* = −0.032, *SE* = 0.010, *p* = 0.001), membership collective self-esteem (*B* = −0.025, *SE* = 0.010, *p* = 0.016), and private collective self-esteem (*B* = −0.035, *SE* = 0.010, *p* < 0.001) were all statistically significant moderators. These negative coefficients demonstrate the buffering effects of collective self-esteem on the internalized racism–anxiety association, meaning that as collective self-esteem increased, the consequences of internalized racism (i.e., anxiety symptoms) were reduced.

For AA participants, the only domain of collective self-esteem that moderates the internalized racism and depression relationship was membership collective self-esteem (*B* = −0.038, *SE* = 0.021, *p* = 0.037). This negative coefficient demonstrates the buffering effect of membership collective self-esteem on the internalized racism–depression association, meaning that as membership collective self-esteem increased, the consequences of internalized racism (i.e., depression symptoms) were reduced. Additionally, no domain of collective self-esteem moderated the relationship between internalized racism and anxiety in Black participants. This suggests that for Black participants, no domain of collective self-esteem buffered the internalized racism–anxiety relation.

For AAPI participants, no domain of collective self-esteem moderated the relationship between internalized racism and depression. Additionally, no domain of collective self-esteem moderated the relationship between internalized racism and anxiety in Asian participants. These findings suggest that for AAPI participants, collective self-esteem did not have a buffering effect on the internalized racism–psychological distress relation.

For Latinx participants, the relationship between internalized racism and depression was moderated by the private collective self-esteem (*B* = −0.097, *SE* = 0.034, *p* = 0.004) and membership collective self-esteem (*B* = −0.084, *SE* = 0.031, *p* = 0.007) domains, which were statistically significant moderators. These negative coefficients demonstrate the buffering effects of collective self-esteem on the internalized racism–depression association, meaning that as collective self-esteem increased, the consequences of internalized racism (i.e., depressive symptoms) were reduced for Latinx individuals. Regarding the relationship between internalized racism and anxiety in Hispanic participants, the private collective self-esteem (*B* = −0.137, *SE* = 0.029, *p* < 0.001) and membership collective self-esteem (*B* = −0.089, *SE* = 0.028, *p* = 0.001) domains were statistically significant moderators. These negative coefficients demonstrate the buffering effects of collective self-esteem on the internalized racism–anxiety association, meaning that as collective self-esteem increased, the consequences of internalized racism (i.e., anxiety symptoms) were reduced for Latinx individuals.

Finally, for Arab/MENA participants, the private collective self-esteem (*B* = −0.081, *SE* = 0.046, *p* = 0.043) and importance to identity (*B* = −0.084, *SE* = 0.051, *p* = 0.037) domains of collective self-esteem were statistically significant moderators of the internalized racism and depression relationship. These negative coefficients demonstrate the buffering effect of collective self-esteem on the internalized racism–depression association, meaning that as collective self-esteem increased, the consequences of internalized racism (i.e., depression symptoms) were reduced for Arab/MENA individuals. Regarding the internalized racism and anxiety relationship, the public collective self-esteem (*B* = 0.124, *SE* = 0.060, *p* = 0.027) was a statistically significant moderator. This negative coefficient demonstrates the buffering effect of public collective self-esteem on the internalized racism–anxiety association, meaning that as public collective self-esteem increased, the consequences of internalized racism (i.e., anxiety symptoms) were reduced for Arab/MENA individuals.

## 4. Discussion

The purpose of the present study was to examine the relationship between internalized racism and depression and anxiety for ethnoracial groups and evaluate the potential moderating effects of the domains of collective racial self-esteem. As a derivative of colonial mentality or internalized colonialism, internalized racism is a construct known to be associated with psychological distress for racial minority populations [4,24,70]. Much of the research about the adverse effects of internalized racism is on African Americans (AAs; 4), and thus, we aimed to investigate this phenomenon with a racially diverse sample. The current study will be one of the few to demonstrate the negative effects of internalized racism on Arab/MENA people while adding to existing knowledge about the effects on AA, AAPI, and Latinx people. We intend to build on the extant literature with this investigation of the impacts of internalized racism with a diverse sample.

Among all participants, the results suggested a positive direct relationship between internalized racism and depression in that increases in reported internalized racism contributed to increases in reported symptoms of depression. Likewise, the direct relationship between internalized racism and anxiety was also positive, indicating that increased endorsement of internalized racism contributed to elevated symptoms of anxiety. However, this direct relationship was not statistically significant for participants who identified as Arab. These findings align with the existing literature, which suggests that people who identify as racial or ethnic minorities who experience internalized racism also experience elevated levels of psychological distress [20,21,71]. Psychologists and other health service professionals must be able to recognize how internalized racism manifests in their clients, be cognizant of the emotional and psychological effects of internalized racism, and create safe, affirming, and nonjudgmental spaces where the client can be vulnerable and share their race-related experiences.

The findings support our first hypothesis that endorsement of internalized racism would relate to elevated symptoms of depression for AA, Latinx, and AAPI populations. To our knowledge, our findings are one of the first to ascertain the effects of internalized racism and depression for people who identify as Arab/MENA. With this knowledge it is important for psychologists and other practitioners to (a) increase critical consciousness among Arab/MENA individuals about the impacts of internalized racism on their mental health, (b) identify and evaluate the ways internalized racism manifests for Arab/MENA people, and (c) obtain a deeper understanding of the historical origins and transmissions of internalized racism across generations in Arab/MENA communities. These actions *should* raise awareness in this historically marginalized identity group of the fact they may be adopting racist beliefs and behaviors from the dominant group to their own detriment. These actions in no shape or form are enough to reduce or eliminate racism but may ameliorate this one specific *consequence* of exposure to and indoctrination by racism.

Additionally, these results are consistent with previous research that indicates acceptance of oppressive messaging and racialized stereotypes, perpetuated by White dominant culture, leads to common symptoms of depression, such as self-degradation, self-loathing or self-devaluation, feelings of hopelessness, poor self-image, or self-destructive behaviors [4,5,7,19]. The propensity to turn inward is commonly observed in individuals suffering from symptoms of depression and often includes depressed mood, feelings of worthlessness, low self-esteem, decreased energy, and difficulty with focus [64]. Thus, when ethnoracial individuals believe and assimilate to Eurocentric ideologies and misconceptions about their own racial group, they unconsciously or consciously engage in isolation, exclusion, and distancing themselves from the very people who could help them [13,14,45].

The findings support our second hypothesis that internalized racism would be associated with symptoms of anxiety. Consistent with previous research, AAs, AAPIs, and Latinx people reported increased anxiety resulting from experiences of internalized racism [8,18]. These findings highlight the ways internalized racism is harmful for racial minority individuals. Unlike depression, the relationship between internalized racism and anxiety among those who identified as Arab/MENA did not reach statistical significance. A possible explanation could be the idiosyncrasies of race as a social construct for Arab/MENA people. As mentioned, people who self-identify as members of this heterogeneous group may not align themselves with the way race is interpreted but prefer an ethnoracial marker that is more salient to their experience [40]. Nonetheless, prejudice, discrimination, stigma, and other intersecting oppressions have deleterious impacts on Arab/MENA people’s livelihoods and quality of life, and thus, a more inclusive construct (e.g., the inclusion of culture, ancestry, and religion) is needed to determine the impact of internalized oppression [39,51].

Our third hypothesis testing for the moderating impacts of collective self-esteem was partially supported. Psychologists and other health service providers must capitalize on their ethnoracial clients’ strengths or other protective factors, such as collective self-esteem, while empowering their clients to share their racial and cultural stories to facilitate healing. A novel finding that extends the literature on the moderating effects of collective self-esteem on the relationship between internalized racism and psychological distress is that the Arab/MENA participants reported reduced depression and anxiety when reporting higher levels of collective self-esteem. Specifically, private collective self-esteem (i.e., how one feels about their social group) and importance to identity (i.e., how important membership in a group is to self-concept) reduced symptoms of depression, and public collective self-esteem (i.e., one’s perceptions of how others feel about one’s social group) reduced symptoms of anxiety [42]. Not only do these findings acknowledge that some Arab/MENA people experience internalized racism, but the results suggest that collective self-esteem could have a buffering effect, protecting Arab/MENA people from the psychological distress deriving from stigmatization, racial discrimination, prejudice, and other oppressions by people in this community [39,41].

Only one domain of collective self-esteem revealed a significant finding for AAs in the sample. Membership collective self-esteem, or perceptions of worth based on membership in the social group, reduced the impacts of internalized racism on depression but had no effect on anxiety [42]. Essentially, those who perceive themselves as a valued member of their racial or social group experience less depression despite the presence of internalized racism. Perhaps having a sense of belongingness versus feeling like a burden to one’s ethnic or racial group are constructs that need to be investigated in future research and may be a better fit in the potential reduction in anxiety as opposed to membership collective self-esteem. Barrie and colleagues (2016 [52]) denoted collective self-esteem as a factor in reducing the effects of both depression and anxiety, but their investigation included a sample of young AA adults. It is possible that there are age differences in how one experiences collective self-esteem that contribute to or detract from the strength of the variable on anxiety.

Consistent with previous research demonstrating the moderating effects of collective self-esteem for Latinx individuals, private and membership collective self-esteem served as buffers and reduced their reported symptoms of depression and anxiety [42,50,51]. These findings provide meaningful information to psychologists or health service professionals working with Latinx communities, as collective self-esteem can serve as a cultural strength and protective factor that may reduce psychological distress. Lastly, the only group that produced a nonsignificant finding for the moderating effects of collective esteem on depression and anxiety was the AAPI participants. Perhaps there are other more impactful protective factors (e.g., collectivism, social support) that are a better fit for reducing the psychological distress deriving internalized racism in this community. From these findings, we can infer that the inclusion of protective factors, such as collective self-esteem, is necessary in the treatment of depression and anxiety deriving from internalized racism among ethnoracial people.

For psychologists, counselors, and therapists, these findings are very important and can inform treatment and intervention decisions when working with clients/patients from historically marginalized populations. Specifically, when considering group-based therapy, improving collective self-esteem improved the effectiveness of the group-based interventions [68]. Taking steps to improve collective self-esteem may be a relatively simple method clinicians can use to improve the effectiveness of group therapeutic interventions with their clients holding marginalized identities. In Asian women college students, longitudinal research indicated that collective self-esteem is a predictor of later life satisfaction, suggesting that clinicians can target the improvement of collective self-esteem to improve the psychological outcomes of their clients [72,73,74,75]. In first-year college students in the United States, improvements in collective self-esteem led to improved psychological and social adjustment as well as improved academic achievement [70]. Rational Emotive Behavior Therapy (REBT) supports the notion that improving collective self-esteem is related to improving psychological functioning and adjustment [75].

### 4.1. Limitations and Future Directions

There are several limitations that should be noted in the present study. First, the present study relied on self-reported survey data to assess all constructs, which is consistent with the existing psychological literature. However, these self-reports are inherently limited as people are often biased, consciously or otherwise, when reporting on their own experiences [76]. Additionally, the present study was cross-sectional in nature; future research should implement a longitudinal approach to examining the associations between internalized racism, psychological distress, and collective self-esteem. The present study, in addition to being cross-sectional, was also not an experiment as there was no manipulation of the independent variable (i.e., internalized racism) or random assignment to condition(s). There are multiple opportunities for improvement and advancement via future research. For example, as the evidence of the effects of internalized racism continues to grow for racially diverse people, research must be more attentive to how this variable is operationalized and its utility for some groups (e.g., Arab/MENA people). Additionally, country of origin/lineage, country of birth, immigration status, and other participant characteristics should be examined to determine their effect, if any, on the relationship between internalized racism and mental health in various contexts.

More evidence of the impacts of internalized racism and intersections with other systems of oppression (discrimination, prejudice, and bias) is also a necessity. In addition to systems of oppression, more research is needed on investigating the impacts of the shifting sociopolitical climate (e.g., the enforcement of policies that ban diversity, equity, and inclusion; assaults on critical race theory; implementation of anti-LGBTQ+ bills), global crises (e.g., the war in Israel and Palestine), and institutions that perpetuate racism and inequities (e.g., education and healthcare). Future research focusing on the evaluation of internalized racism with the intersection of oppressed identities (e.g., gender, race/ethnicity, and sexual) is critical for psychologists to consider. In addition, we must continue to identify and examine protective factors (e.g., social support and collective self-esteem), personal strengths (e.g., resilience and sense of belonging), and risk factors (e.g., isolation and feelings of burdensomeness) on the relationship between internalized racism and psychological and physical health.

The investigation of internalized racism and oppression across time, such as the intergenerational transmission, must be investigated, as colonized mentality, self-degradation, and other components can be passed down through generations due to shared hurt, trauma, and violence. Additionally, we must be conscientious of the impacts of colonization on our methods of investigation including the constructs and variables we create. Research must operate from a lens of decolonization when attempting to understand the impacts of internalized racism or oppression on historically marginalized racial groups to activate liberation and avoid further marginalization. This may include using positive psychological variables (e.g., life satisfaction) as outcomes in psychological research, examining the experiences of racial minority persons from a strengths-based perspective, and/or using qualitative research methods to inform and reform quantitative work. The use of qualitative methods allows researchers to understand the phenomena as experienced by marginalized people through their own words and descriptions. This current study was informed by qualitative work undertaken by the lead author who spoke with 16 people with racial minority identities in focus groups discussing their experiences with media, family, and societal messages about their own worth and the worth of their racial group. This decolonized approach will shift research from a standpoint of “European Rightness” to standpoint of understanding and cooperation with individuals holding marginalized racial identities. Lastly, we want to explore how collective racial self-esteem can connect broadly to resistance and social justice efforts. It has been shown from this study that collective racial self-esteem can have a buffering impact for internalized racism, but understanding how this can advance social justice efforts and decrease resistance with both in- and out-group experiences is yet to be examined. Statistically, in the regression models, the variance inflation factor (VIF) *was not* examined as there were no inflated standard errors (see Table 2 and Table 3) or Pearson’s correlations greater than 0.9 (see Table 4).

### 4.2. Constraints on Generalizability

The present study used a sample of racially diverse minority individuals recruited via the MTurk platform. Although each analysis examined the sample as a whole and then examined the racial groups separately, there were no steps taken by the researchers to examine within-group variance based on sub-groups (e.g., country of origin or immigration status). Additionally, there was an imbalance across racial groups. This imbalance may affect the statistical analyses (i.e., power) and thus affect the results/findings. Future research should aim to have a larger subset of *each* racial/cultural group to avoid limitations based on sample size. Also, the data were gathered during the COVID-19 pandemic and during a time of civil unrest due to nationwide racial justice protests. Finally, although there are some constraints on generality identified, we have no reason to believe that the results depend on other characteristics of the participants, materials, or context.

## 5. Conclusions

Overall, our findings suggested that internalized racism is associated with psychological distress in racial and ethnic minority individuals. Additionally, the findings suggested that specific domains of collective self-esteem weakened the aforementioned association for certain racial groups. Identifying specific constructs that have the potential to weaken the association between internalized racism and psychological distress, then, would be of paramount interest and importance to practicing psychologists and therapists to combat the previously established negative outcomes of internalized racism in members of minority groups.

## Figures and Tables

**Figure 1 behavsci-14-01003-f001:**
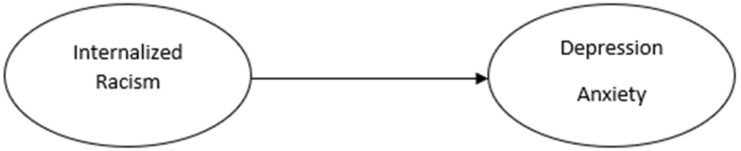
Model of internalized racism and psychological distress.

**Figure 2 behavsci-14-01003-f002:**
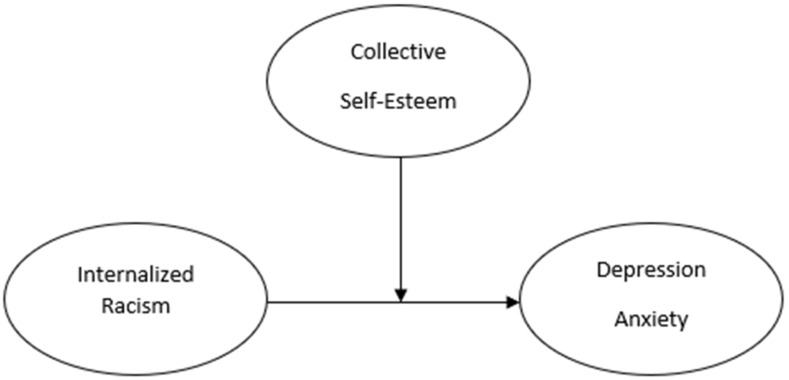
Model of the internalized racism and psychological distress association moderated by collective self-esteem.

**Table 1 behavsci-14-01003-t001:** Participant characteristics.

Gender Identity		N	Percentage
	Male	248	46.2
	Female	283	52.7
	Transgender	4	0.7
	Non-Binary	2	0.4
**Income**		**N**	**Percentage**
	USD 0–20,000	75	13.9
	USD 20,001–40,000	133	24.7
	USD 40,001–60,000	158	29.4
	USD 60,001–80,000	80	14.9
	USD 80,001–100,000	45	8.4
	USD 100,001 and above	46	8.6
**Relationship**		**N**	**Percentage**
	Single	214	39.8
	In a dating/committed relationship	97	18.0
	Married	197	36.9
	Divorced/separated	25	4.7
	Widowed	3	0.6
**Race**		**N**	**Percentage**
	Arab	68	12.6
	Asian	201	37.4
	Black	163	30.3
	Hispanic/Latinx	97	18.0
	Multiracial	1	0.2

**Table 2 behavsci-14-01003-t002:** Standard linear regression of internalized racism on depression.

	All (N = 483)		Black (n = 149)		Asian (n = 184)		Hispanic (n = 84)		Arab (n = 57)	
	*B*	*SE*	*B*	*SE*	*B*	*SE*	*B*	*SE*	*B*	*SE*
AROS	0.582 ***	0.034	0.447 ***	0.070	0.691 ***	0.044	0.386 ***	0.089	0.317 **	0.130
Age	0.011	0.044	−0.042	0.084	−0.065	0.057	0.230	0.107	−0.142	0.192
Gndr	0.063	0.039	−0.129	0.075	0.030	0.059	0.230	0.087	0.034	0.136
Rtshp	−0.120 **	0.044	−0.125	0.093	0.000	0.067	−0.301	0.104	−0.062	0.187
Incm	−0.089	0.038	−0.125	0.073	0.015	0.059	−0.152	0.089	−0.122	0.121

***Note*.** AROS = internalized racism; Gndr = gender; Rtshp = relationship status; Incm = income. ** *p* < 0.01, *** *p* < 0.001.

**Table 3 behavsci-14-01003-t003:** Standard linear regression of internalized racism on anxiety.

	All (N = 481)		Black (n = 151)		Asian (n = 182)		Hispanic (n = 82)		Arab (n = 57)	
	*B*	*SE*	*B*	*SE*	*B*	*SE*	*B*	*SE*	*B*	*SE*
AROS	0.621 ***	0.034	0.537 ***	0.070	0.710 ***	0.044	0.420 ***	0.089	0.258	0.130
Age	0.011	0.044	−0.042	0.084	−0.065	0.057	0.230	0.107	−0.142	0.192
Gndr	0.063	0.039	−0.129	0.075	0.030	0.059	0.230	0.087	0.034	0.136
Rtshp	−0.120 **	0.044	−0.125	0.093	0.000	0.067	−0.301	0.104	−0.062	0.187
Incm	−0.089	0.038	−0.125	0.073	0.015	0.059	−0.152	0.089	−0.122	0.121

***Note*.** AROS = internalized racism; Gndr = gender; Rtshp = relationship status; Incm = income. ** *p* < 0.01, *** *p* < 0.001.

**Table 4 behavsci-14-01003-t004:** Study variable Pearson correlations.

	1	2	3	4	5	6	7
1. AROS	-						
2. Dep	0.50 ***	-					
3. Anx	0.52 ***	0.78 ***	-				
4. CseI	−0.42 ***	−0.22 ***	−0.22 ***	-			
5. CseM	−0.57 ***	−0.52 ***	−0.40 ***	0.42 ***	-		
6. CsePr	−0.75 ***	−0.46 ***	−0.45 ***	0.44 ***	0.68 ***	-	
7. CsePu	−0.09 *	−0.22 ***	−0.20 ***	0.06	0.13 **	0.13 **	-

***Note*.** AROS = internalized racism; Dep = depression; Anx = anxiety; CseI = collective self-esteem importance to identity; CseM = collective self-esteem membership esteem, CsePr = collective self-esteem private regard; CsePu = collective self-esteem public regard. * *p* < 0.05, ** *p* < 0.01, *** *p* < 0.001.

## Data Availability

The data are available upon reasonable request from the corresponding author.

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
