# Peer review of "Internalized Racism and Mental Health: The Moderating Role of Collective Racial Self-Esteem"

_behavsci, 2024, doi:10.3390/bs14111003_

Round 1
Reviewer 1 Report
Comments and Suggestions for Authors
Thank you for the opportunity to review this interesting and well-written manuscript. Incorporating the concept of collective racial self-esteem into the association between internalised racism and mental health outcomes provides valuable insights in how to mitigate the harms of internalised racism and improve the wellbeing of racialised people.
I have a few comments and editing suggestions.
1. I think it would be helpful to explain why investigators chose to investigate internalised racism exclusively and about the specific characteristics of internalised racism when compared with other manifestations.
2. I think in studies about racism which are recruiting people on the basis on of their experiences of racialisation should include a positionality statement about the research team and the conceptualisation of racism as collectively understood by the team.
3. The strength of the correlation between racial identity and the groupings used in the study is unclear, e.g. is MELA a racial identity or an administrative category, do people identify with AAPI enough to feel collective racial self esteem within that grouping? I think the usage of these groupings needs to be supported by literature.
4. Some additional methodology information that should be included:
-Did the demographic questionnaire have an open-ended option or allow participants to select multiple racial categories?
-Was the study available in languages other than English
-What was the date on which the study was conducted?
-Does the DASS-21 questionnaire have cut-offs for high scores that indicate depression/anxiety/ psychological distress or is a score of zero what would be expected of someone who was experiencing good wellbeing?
5. Table 2 was not referred to in the text and column labels were missing. Dep and Anx were missing from the note explaining the acronyms used in the table.
6. Line 440: the word racism is missing after internalized
7. Line 447-452: To me, this framing may risk problematising internalised racism and positing increasing critical consciousness among MENA individuals as a solution rather than racism and racialisation in society as the problem and the elimination of racism as the solution. If this is not the intention, then clarifying the scope of these proposals to the context of an individual therapeutic relationship or extending the scope of the learning points beyond what is currently written may be useful.
Author Response
Thank you so much on behalf of my coauthors for your thoughtful review of our work. We believe your review strengthened our manuscript. Please find our responses to your review in the attached document.

Reviewer 2 Report
Comments and Suggestions for Authors
I consider this study to be a very interesting venture as it gives more insight into relations between internalised racism and depression/anxiety measures (using other than classic Beck's Inventory tools is a merit of the study). The research is well designed the sample male-female ratio is good (which should be obvious but in many studies I have reviewed so far is not so methodologically obvious to take that into consideration). The idea of collective self-esteem is also interesting. I see many paralels with studies on LGBT population and internalised homophobia maybe give some reference to that? Just a suggestion. In more homophobic societies (like Eastern European countries or the Bible Belt in the US) collective self-esteem of LGBT population may be lower - it would be interesting to pursuit those lets say environmental/cultural factors of the sample. Otherwise the study is well structured and carefully prepared with all proper sections eg limitations of the study as well as ethical concerns (like obtaining informed consent and so on) so I recommend it for publication in the present form.
Author Response
Comment 1: I consider this study to be a very interesting venture as it gives more insight into relations between internalised racism and depression/anxiety measures (using other than classic Beck's Inventory tools is a merit of the study). The research is well designed the sample male-female ratio is good (which should be obvious but in many studies I have reviewed so far is not so methodologically obvious to take that into consideration). The idea of collective self-esteem is also interesting. I see many paralels with studies on LGBT population and internalised homophobia maybe give some reference to that? Just a suggestion. In more homophobic societies (like Eastern European countries or the Bible Belt in the US) collective self-esteem of LGBT population may be lower - it would be interesting to pursuit those lets say environmental/cultural factors of the sample. Otherwise the study is well structured and carefully prepared with all proper sections eg limitations of the study as well as ethical concerns (like obtaining informed consent and so on) so I recommend it for publication in the present form.
Response 1: Thank you for your feedback and for agreeing to review our manuscript! We didn't mention it here, but our lab has some research on internalized homophobia. I didn't see a natural way to reference this pending work. I agree that it is important, and we are interested in looking at this phenomenon cross-culturally as well.
Reviewer 3 Report
Comments and Suggestions for Authors
This manuscript offers a comprehensive and insightful examination of how internalized racism impacts the mental health of ethnic minority groups, with a particular focus on the moderating role of collective racial self-esteem. The inclusion of various populations, such as African Americans, Asian Americans/Pacific Islanders (AAPI), Latinx Americans, and Arab/Middle Eastern/North African (AMENA) individuals, underscores the relevance of this study against the backdrop of increasing multiculturalism. The authors have made a significant contribution to the growing body of literature that addresses the psychological consequences of internalized racism. The research methodology, which utilizes the Hayes PROCESS macro for moderation analysis, is well-suited to the study's research question. The authors’ focus on collective racial self-esteem as a protective factor adds an important dimension to understanding psychological resilience mechanisms, which I find highly relevant.
The theoretical foundation is well anchored in existing literature on internalized racism and its effects on mental health. The authors skillfully integrate references to established frameworks, such as the collective self-esteem theory developed by Luhtanen and Crocker, and more recent empirical work that supports the moderating effects of collective racial self-esteem. The manuscript portrays internalized racism as an insidious form of oppression that can lead to psychological issues, in alignment with previous research findings. Moreover, the theoretical foundation in both racial identity research and mental health studies is well-established, and the data presented support the hypotheses proposed by the authors.
Although the manuscript covers internalized racism well from the literature review, it could benefit from further integration of global perspectives on internalized oppression, particularly incorporating research on postcolonial studies and the impacts of systemic racism in non-Western contexts. While colonialism is often used as a term related to oppression, it is not clearly defined or distinguished from oppression in this context. The manuscript briefly touches on the interaction between internalized racism and other systems of oppression, but this could be enhanced by incorporating an intersectional perspective that considers how internalized racism interacts with factors such as gender, class, and immigration status. Particularly for the respondents from the MENA region, it would be highly relevant to investigate whether the facet of religious identity (Christian Arabs or Muslim Arabs) has a significant impact. This would allow for a more holistic view of how various forms of marginalization exacerbate mental health outcomes.
While collective racial self-esteem is posited as a protective factor, cultural differences in how collective identity is perceived by different groups could influence the findings. For instance, community values in Latinx and Asian American communities could function differently as buffers against internalized racism. Delving deeper into these cultural nuances could enrich the discussion section and provide better-tailored mental health interventions. It would also be important to know whether Brazil was included in the Latinx group or if it was limited to Spanish-speaking populations. In this context, the edited volume by Vasquez and Rocha on the Brazilian diaspora in the U.S., which highlights important identity markers, is particularly noteworthy.
The graphs and tables presented in the manuscript appear clear and well-labeled, but there are a few points to note. The sample size (N = 526) is fairly balanced, but the discrepancy in representation across racial groups (e.g., only 68 Arab/AMENA participants) may affect the generalizability of the findings. The authors should highlight the limitations of unequal representation in certain analyses.
Regarding the correlation table (Table 2), the reported correlations between internalized racism and psychological stress variables (e.g., depression and anxiety) are robust (e.g., AROS and Depression, r = .50***), but further clarification of the practical significance of these moderate to strong relationships could enhance the discussion. Additionally, a discussion of whether collinearity was tested among these variables would assure readers of statistical rigor.
The practical applications of these findings for mental health professionals are significant, but the authors could provide more concrete examples of how practitioners might use these insights in therapy, counseling, or community interventions, particularly by leveraging collective racial self-esteem as a therapeutic tool.
Overall, however, this manuscript is a very well-executed study that makes a valuable contribution to the literature on internalized racism and mental health. While there are areas that could be expanded or clarified, particularly regarding the cultural nuances of collective racial self-esteem, the research is solid and methodologically sound. The authors are encouraged to continue this important work and further explore the intersections of race, culture, and mental health in future research endeavors.
Author Response
Thank you for your in-depth review of our manuscript. We responded to your thoughtful comments and feedback in the attached document. On behalf of myself and my coauthors, you have our gratitude.
